# Detection and Molecular Characterization of Novel Porcine Parvovirus 8 Strains in China

**DOI:** 10.3390/v17040543

**Published:** 2025-04-08

**Authors:** Wei Chen, Yanqing Hu, Yan Qin, Yuying Li, Xinyu Zhang, Haixin Huang, Mengjia Liu, Yuping Zheng, Xuelian Lu, Qiaoqiong Wang, Jianuo Yang, Liting Kang, Lulu Xie, Baopeng Zhao, Tian Lan, Wenchao Sun

**Affiliations:** Wenzhou Key Laboratory for Virology and Immunology, Institute of Virology, Wenzhou University, Wenzhou 325035, China; 23451039005@stu.wzu.edu.cn (W.C.);

**Keywords:** PPV8, phylogenetic analysis, homology, genome, parvovirus

## Abstract

Porcine parvovirus 8 (PPV8), the most recently discovered PPV genotype, was first reported in pigs in Guangdong, China, in 2021. In this study, we assessed 69 lung tissue samples collected from animals with high fever or respiratory syndrome on pig farms in Guangxi in 2018. Five nearly full-length genome sequences were characterized and analyzed. The percentage of PPV8-positive samples was 13.04% (9/69), and five complete PPV8 sequences were obtained, which were designated PPV8-A, PPV-B, PPV8-G, PPV8-H, and PPV8-I. The genomic sequence homology among the five PPV8 strains and other PPVs was 25.91–98.84%, with the closest genetic relationship to PPV8-GDJM2021 (98.84%), followed by PPV1 (44.64%). For the NS1 protein, phylogenetic analysis revealed that the identified PPV8-I was closely related to PPV8-GDJM2021 and that PPV8-A was closely related to PPV8-H, whereas PPV8-B and PPV8-G were more distantly related to the other PPV8 strains. For the VP1 protein, phylogenetic analysis revealed a close correlation between PPV8-H and PPV8-GMJM2021, whereas PPV8-A, PPV8-B, PPV8-G, and PPV8-I were more distantly related. In conclusion, five nearly full-length sequences were amplified, and the molecular characteristics of PPV8 were analyzed. These findings improve our understanding of the PPV8 genome.

## 1. Introduction

Parvovirus is a member of the *Parvoviridae* family and is a small, flexible, nonenveloped, negative single-stranded linear DNA virus [1,2]. *Parvoviridae* contains three subfamilies: *Parvovirinae* and *Hamaparvovirinae,* which include viruses that infect vertebrate (including human) hosts, and *Densovirinae*, which includes viruses that infect invertebrate (including insect and arthropod) hosts [3,4]; viruses that infect pigs are of the *Parvovirinae* and *Hamaparvovirinae* subfamilies [5]. Currently, there are 9 parvovirus genera, of which *Protoparvovirus*, *Teraparovirus*, *Chapparvovirus*, *Copiparvovirus*, and *Bocaparvovirus* infect swine [6,7]. The porcine parvovirus genome is approximately [4,5,6] kb long and contains two major open reading frames, ORF1 and ORF2, which encode nonstructural (NS) proteins and viral capsid (VP) proteins, respectively (Figure 1). The NS proteins play an important role in viral replication, especially in DNA replication, whereas the VP proteins are related to the transcription and translation of the parvovirus genomes [8,9]. In addition, a 5′-untranslated region (UTR) and a 3′-untranslated region (UTR) exist at the ends of the parvovirus genome, which usually form complex palindromic hairpin structures of approximately 120–200 bases [10].

To date, eight genotypes of porcine parvovirus have been identified (PPV1–PPV8), and these have been designated as novel PPVs (nPPVs). Porcine parvovirus 1 (PPV1), which is one of the major causes of reproductive pig disease, was first isolated in Germany in 1965 [11,12]. PPV1 belongs to the genus Protoparvovirus and has been shown to be responsible for stillbirth, mummification, embryonic death, and infertility [7,13]. During the last two decades, several novel parvoviruses have been identified in pigs by molecular methods and have been designated as PPV2–PPV8. PPV2 was discovered in 2001 in Myanmar and was proposed as the primary pathogen of the porcine respiratory disease complex (PRDC) [14,15]. April Nelson et al. reported that the accumulation of PPV2 nucleic acid in the cytoplasm of alveolar macrophages may be due to the phagocytosis of viruses during viremia or active PPV2 replication [16]. PPV3, also known as porcine PARV4 and porcine Hoko virus and closely related to human parvovirus 4 (PARV4), was identified in Hong Kong in 2008 [17]. These viruses all belong to the *Tetraparvovirus* genus. Three PPVs, PPV4, PPV5, and PPV6, have been identified and belong to the genus *Copiparvovirus* [10]. PPV7 was discovered in the United States in 2016 and belongs to the genus *Chaphamaparvovirus* [18]. To date, the specific pathogenic mechanism of the PPV3–PPV8 viruses is unknown, and further research is needed. However, at present, many studies have reported the coinfection of PPVs with other disease-causing pathogens. Komina et al. reported signs of apoptosis and necrosis in cell cultures of SPEV and SK cells infected with PPV6, indicating that the pathogenic mechanism of PPV6 may be closely related to these processes [19]. In addition, PPV1 and PPV6 showed a clear coinfection association with PCV2-associated diseases (PCVADs) regardless of PRRSV infection, and PPV2 and PPV7 showed a clear coinfection association with PRRSV regardless of PCVAD [20]. In addition, Jinhui Mai et al. speculated that PPV7 is an important cofactor of PCV3 that can stimulate the replication of PCV3 and increase its viral load [21]. These findings indicate that porcine parvovirus has a significant effect on the reproduction and breeding of swine, making research on porcine parvovirus particularly important.

In this study, we detected the presence of PPV8 in lung tissue samples from diseased pigs with detection primers, with 9 out of 69 samples showing positivity. Five of the PPV8 strains were sequenced and designated PPV8-A, PPV8-B, PPV8-G, PPV8-H, and PPV8-I. In this experiment, we generated near-complete genome data for five different PPV8 strains and performed sequence alignment and phylogenetic tree analysis, laying the foundation for future research on PPV8.

## 2. Materials and Methods

### 2.1. Clinical Samples

In this experiment, we collected 69 lung tissue samples from pigs with high fever or respiratory syndrome on pig farms in Guangxi Province in 2018, homogenized them, and stored them at −80 °C.

### 2.2. Sample Processing

To investigate the presence of DNA viruses in pigs with high fever or respiratory syndrome, 1.0 g of the 69 lung tissue samples were weighed into 1.5 mL centrifuge tubes, and 300 µL of PBS was added to each tube for homogenization. The tubes were centrifuged at 12,000 rpm for 10 min at 4 °C, and two-thirds of the supernatant was collected and filtered through a 0.45 µm filter [22,23]. We subsequently used a TIANamp virus DNA/RNA Fast Kit (TIANGEN, Beijing, China) to extract total DNA. Next, the extracted DNA was subjected to PCR. In addition, we filtered and sterilized the remaining supernatant with a 0.22 µm filter and stored it at −80 °C for virus isolation.

### 2.3. PCR Detection of PPV8 and Sequencing

To determine the prevalence of PPV8 in pig farms in Guangxi, the nest-PCR (nPCR) method was used to detect PPV8-positive samples among the 69 clinical samples. A total of 9 positive samples were detected, 5 of which were further characterized. PPV8-specific primers were designed on the basis of the conserved sequence of the PPV8-GDJM2021 strain (GenBank: OP021638.1) [24]. PCR amplification was performed with the outer detection primer pair PPV8-outF/R and the inner detection primer pair PPV8-inF/R. The thermal cycling conditions involved 35 cycles of denaturation, annealing, and extension for nested PCR (Table 1). We sent the positive samples to Sangon Biotech (Shanghai, China) for sequencing and removed the false positive samples through BLAST 1.4.0 identification. Four pairs of primers were subsequently designed on the basis of conserved regions of the nearly full-length sequence of PPV8-GDJM2021 to amplify four DNA fragments [25]. The specific annealing temperatures for each pair of primers and the primer sequences are shown in Table 1. nPCR amplification was performed under the same conditions. The PCR products were analyzed on a 1% agarose gel containing Tris-acetate buffer solution (pH 8.0) and Gelstain (TransGen Biotech, Beijing, China) [26]. We cloned the target fragment into the pMD™18-T vector (TaKaRa, Hong Kong, China) [26] and sent the positive clones to a bioinformatics company for sequencing. These sequences were then assembled with the SeqMan program in the DNASTAR package [27]. Afterward, the splicing results were analyzed for amino acid and nucleotide sequences.

### 2.4. Sequence Alignment and Phylogenetic Analysis

Sequence comparison was conducted with MEGA 11 to analyze the sequence differences of nucleotides (nt) and amino acids (aa) between PPV8-GDJM2021 and the five DNA sequences assessed in this analysis. A maximum likelihood tree (MLT) model (T92+G) with the Tamura–Nei model was constructed with MEGA 11 software [28]. Bootstrap analysis (1000 replicates) was used to construct a phylogenetic tree [29]. We then used the Evolview optimization tool to construct the evolutionary tree [30]. In addition to those of PPV1–PPV8, the nucleic acid sequences encoding the NS1 protein of several other parvoviruses were obtained from GenBank.

### 2.5. Nucleotide Sequence Accession Numbers

The obtained sequences were assembled into full genomes with Lasergene software 17.3.1 and deposited in the NCBI GenBank database under the following accession numbers: PP842644–PP842648. For genetic analysis, *Parvoviridae* family genomes were retrieved from GenBank. The general information and accession numbers of all 46 porcine parvovirus strains are listed in Table A1.

## 3. Results

### 3.1. Genomic and Amino Acid Structure Analysis of PPV8

The near-full-length genome of PPV8 contained 4211~4215 bases with a G + C content ranging from 36.14 to 36.74%. The genome sizes are expected to be larger, and sequencing of the ends may have been hampered by palindromic structures 10. Restriction enzyme site analysis was performed with DNAMAN 10.0 software, and strains A, B, G, H, and I had 66, 63, 70, 57, and 62 restriction enzyme sites, respectively.

Putative ORFs were obtained with the NCBI ORF Finder tool and then identified by protein BLAST 1.4.0 analysis with the NCBI RefSeq database [31]. ORF1 is 1806 nt long and encodes a putative nonstructural protein (NSP) of 601 amino acids (aa), and ORF2 is 2106 nt long and encodes a putative capsid protein (VP) of 701 amino acids (aa) (Figure 1). ORF1 and ORF2 overlap by 14 nt, which is consistent with the reported PPV8-GDJM2021 sequence and inconsistent with the ORFs of other PPVs (PPV1–PPV7) [32]. The genomes of the five PPV8 strains in this research contained a 5′-untranslated region (UTR) of more than 200 nt and a 3′-untranslated region (UTR) of more than 191 nt at both ends.

The amino acid sequences of the five PPV8 strains identified in this study were subsequently analyzed. The sequences of the five strains were aligned pairwise, and multiple sequences were aligned with PPV8 with DNAMAN and MEGA 11 software.

Pairwise comparisons were performed for the nucleotide sequences and predicted amino acid sequences of the five PPV8 strains with those of other PPVs. The results revealed that the genomes of the five PPV8 strains shared 25.91–44.64% DNA sequence identity with those of other members of Parvovirinae (PPV1–PPV7) and were most closely related to those of PPV1 (Table A1). Within NS1 (VP1), the five strains presented 98.84–99.67% (98.57–99.43%) and 98.67–99% (97.69–98.53%) sequence homology with the PPV8-GDJM2021 strain at the amino acid and nucleotide levels, respectively.

A comparison of the NS1 sequences of the five PPV8 strains with those of PPV8-GDJM2021, PPV1, PPV2, and PPV3 revealed that they all contain conserved ATP- or GTP-binding Walker A loop (GxxxxGKT/S; GPTSTGKS), Walker B (xxxxEE; LGWFEE), Walker B’ (KxxxxGxxxxxxxK; KALTSGQNIRVDQK), and Walker C (PIxIXXN; PILITSN) aa motifs [33]. In addition, the NS1 protein contains two conserved replication initiator (endonuclease) motifs, xxHuHxxxx (GLHFHVLLW) and YxxxK (YFLKK) (conserved amino acids are indicated at the bottom of the alignment, and “u” indicates a hydrophobic residue) (Figure 2). Similarly, the VP1 protein also has a similar conserved domain. A comparison of the VP1 sequences of the five PPV8 strains with those of other PPVs revealed that they all have a conserved motif in the catalytic center (DxxAxxHDxxY; DAAARKHDIAY) of the putative phospholipase A2 domain and that the five PPV8 strains contain a calcium-binding loop (YLGPF) rather than the “YxGxG” motif found in most PPVs [34] (Figure 3).

### 3.2. Phylogenetic Analysis of PPV8

A phylogenetic analysis of 51 PPV strains selected from GenBank and the 5 PPV8 strains identified in this investigation was conducted based on the nucleotide sequences of the NS1 protein and the VP1 protein (Table A2). As shown in Figure 4, for the NS1 protein, the five PPV8 strains identified in this analysis clustered with PPV8-GDJM2021 and belonged to the genus *Protoparvovirus* but branched from other *Protoparvovirus* strains. In addition, PPV8-I was closely related to PPV8-GDJM2021, and PPV8-A was closely related to PPV8-H. PPV8-B and PPV8-G were closely related to the other PPV8 strains. As shown in Figure 5, the phylogenetic tree of VP1 indicated a close correlation between PPV8-H and PPV8-GMJM2021, whereas PPV8-A, PPV8-B, PPV8-G, and PPV8-I were more distantly related.

## 4. Discussion and Conclusions

In recent years, increasing attention and research have focused on DNA viruses that infect pigs, especially porcine parvovirus. With the development of sequencing technology, eight types of porcine parvoviruses (PPV1–PPV8) have been identified in swine populations, among which PPV2–PPV8 are known as new PPVs (nPPVs) [15]. On the basis of the homology of the NS1 protein sequence, parvoviruses are classified as *Protoparvovirus* (PPV1, PPV8), *Tetraparvovirus* (PPV2–PPV3), *Copiparvovirus* (PPV4–PPV6), or *Chapparvovirus* (PPV7) [6,35,36]. Among these, the most recently reported PPV8 was initially identified by high-throughput sequencing (HTS) in porcine reproductive and respiratory syndrome virus (PRRSV)-positive samples collected from swine herds in Guangdong Province in 2021. PPV8 shares 31.86–32.68% aa sequence identity with the NS1 protein of PPV1 and *porcine bufavirus* (PBuV) and represents a new species within the *Protoparvovirus* genus [32].

The newly discovered PPV8 and PPV1 strains belong to the same *Protoparvovirus* genus. PPV8 was discovered four years ago. However, there are currently few reports on the PPV8 strain. To fill this knowledge gap, we conducted genome detection and identification on 69 lung tissue samples from sick swine from a pig farm in Guangxi. The PPV8-positive rate was 13.04% (9/69). In this review, we reported the near-complete genome sequences of five strains (PPV8-A, B, G, H, and I). These strains were identified by BLASTN and were found to have 97.48–97.81% similarity to PPV8-GDJM2021. For the NS1 gene, the five PPV8 strains presented 98.84–99.67% and 98.67–99% sequence homology with the PPV8-GDJM2021 strain at the amino acid and nucleotide levels, respectively. We also conducted a phylogenetic tree analysis of the five strains with other members of the *Parvoviridae* family and reported that PPV8-I was closely related to PPV8-GDJM2021 and that PPV8-A was closely related to PPV8-H, whereas PPV8-B and PPV8-G were more distantly related to the other PPV8 strains.

In conclusion, five nearly full-length sequences were amplified, and the molecular characteristics of PPV8 were analyzed. These findings improve our understanding of the PPV8 genome. Although we have reported five strains of PPV8, their biological characteristics and relationships with diseases are still not fully understood. Further research on PPV8 in the future will help address these questions.

## Figures and Tables

**Figure 1 viruses-17-00543-f001:**
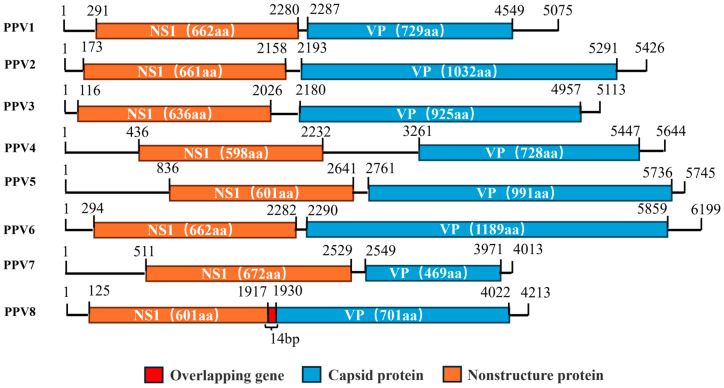
Genomic structures of eight species of porcine parvovirus.

**Figure 2 viruses-17-00543-f002:**
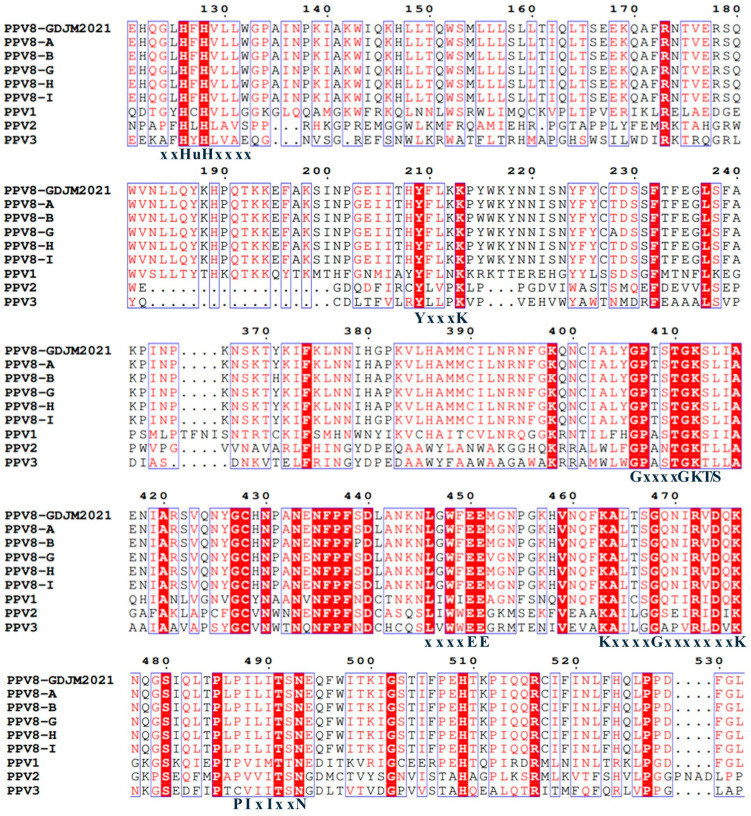
Multiple sequence alignment of the NS1 motif of five PPV8 strains with those of other PPVs. The conserved motifs in the helicase domain of the NS1 proteins and ATP- or GTP-binding regions including xxHuHxxxx (GLHFHVLLW), YxxxK (YFLKK), and Walker motifs (A, B, B’, and C).

**Figure 3 viruses-17-00543-f003:**
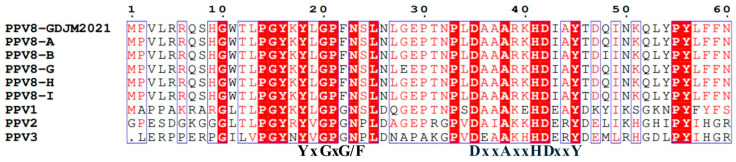
Multiple sequence alignment of the VP1 motif of five PPV8 strains with those of other PPVs. Conserved motifs in the phospholipase A2 (PLA-2) domain and calcium-binding domain (YxGxG) in the VP1 proteins of five PPV8 strains and other PPVs.

**Figure 4 viruses-17-00543-f004:**
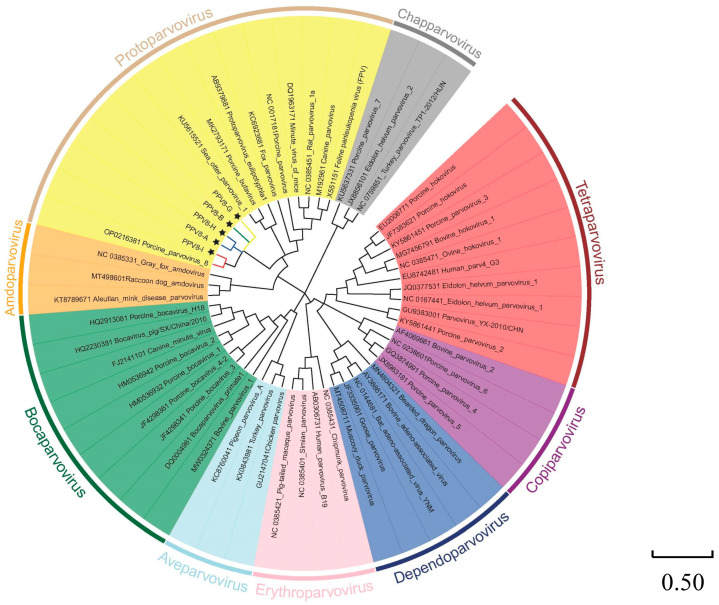
Phylogenetic trees of parvoviruses based on the NS1 protein. Phylogenetic analysis was performed on the basis of the full-length nucleotide sequences of the NS1 protein of PPV8 and other reference strains of nine parvovirus genera retrieved from GenBank by MEGA 11 with the maximum likelihood (ML) method and 1000 bootstrap replicates.

**Figure 5 viruses-17-00543-f005:**
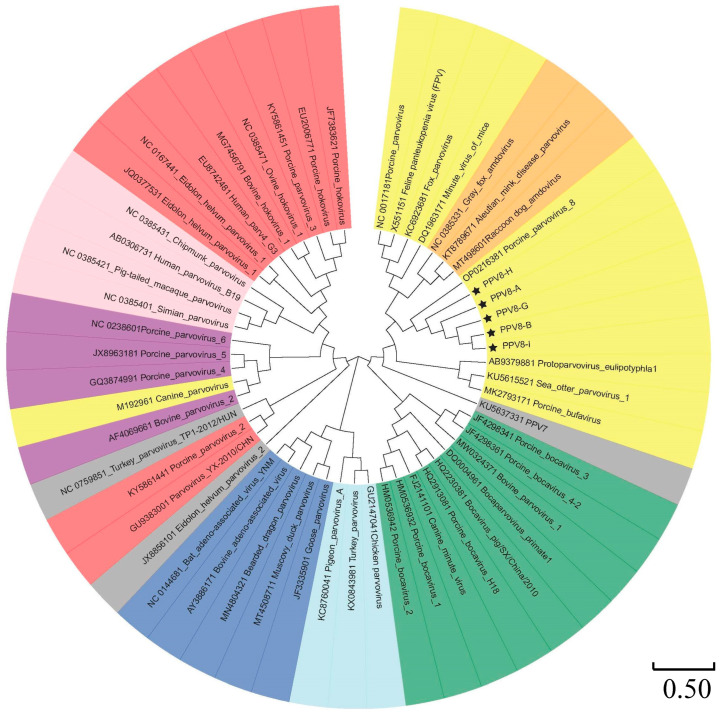
Phylogenetic trees of parvoviruses based on the VP1 protein. Phylogenetic analysis was performed on the basis of the full-length nucleotide sequences of the VP1 protein of PPV8 and other reference strains of nine parvovirus genera retrieved from GenBank by MEGA 11 with the maximum likelihood (ML) method and 1000 bootstrap replicates. The colors corresponding to the VP1 sequence in this figure are consistent with those used in Figure 4.

**Table 1 viruses-17-00543-t001:** Details of the primers and annealing temperatures used in this study.

Primer		Sequence (5′→3′)	PCR Conditions		
			Predenaturation	Denaturation, Annealing, Extension	Final Extension
PPV8	OutF	TGTTGGTTTGCACCTAGCG	98 °C 30 s	98 °C 10 s, 56 °C 15 s, 72 °C 15 s	72 °C 2 min
	OutR	TGATGAGATGGTGGAACGC			
PPV8	InF	TCCAAGTTGCCCTAGACAGC	98 °C 30 s	98 °C 10 s, 56 °C 15 s, 72 °C 15 s	72 °C 2 min
	InR	GCCTCGTACATGTGGACCTC			
①	119F	GAAGAAGAATCTGATTAAGGTAAGCC	94 °C 5 min	94 °C 30 s, 56 °C 30 s, 72 °C 30 s	72 °C 7 min
	712R	GGATAGTTAATAGTGATAGAAGGAGC			
②	684F	TTACTAACTCAATGGTCAATGCTCCTTC	94 °C 5 min	94 °C 30 s, 52 °C 30 s, 72 °C 30 s	72 °C 7 min
	2037R	GATTGTCTTCTAAGGACTGGC			
③	1960F	CCAATACAGACTCACATCTTCTCTTGA	94 °C 5 min	94 °C 30 s, 60 °C 30 s, 72 °C 30 s	72 °C 7 min
	3541R	TGGTTTGTTGTGACATCTCTGCTTCTAA			
④	3488F	CTCATCATCCAAGAGAAGCTC	94 °C 5 min	94 °C 30 s, 55 °C 30 s, 72 °C 30 s	72 °C 7 min
	4311R	ACCCAAGAGCGTTTTCAAAGA			

## Data Availability

Whole-genome sequences of PPV8 discovered in this project are available in GenBank under accession nos. PP842644–PP842648.

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
