# Peer review of "Detection and Molecular Characterization of Novel Porcine Parvovirus 8 Strains in China"

_viruses, 2025, doi:10.3390/v17040543_

Round 1

Reviewer 1 Report (Previous Reviewer 2)

Comments and Suggestions for Authors

Thank you for your thorough revision in response to the reviewer's feedback. The manuscript's quality has greatly improved. 

Author Response

Reviewer 2 Report (Previous Reviewer 3)

Comments and Suggestions for Authors

The authors complied with the severe inconsistencies in the Discussion mentioned by this reviewer which is appreciated. Unfortunately, it remains elusive, how such grave inconsistencies can occur after careful prove reading before a manuscript is submitted.

Round 2

Reviewer 2 Report (Previous Reviewer 3)

Comments and Suggestions for Authors

Fine with me.

This manuscript is a resubmission of an earlier submission. The following is a list of the peer review reports and author responses from that submission.

Round 1

Reviewer 1 Report

Comments and Suggestions for Authors

The manuscript presents a study on the detection and molecular characterization of porcine parvovirus 8 (PPV8) strains in China. This is not the first report of PPV8 identification in China, but the study may contribute to the growing understanding of porcine parvoviruses and their genetic relationships. The research includes extensive sequence analysis and provides valuable genomic data. However, several areas require improvement, particularly in terms of clarity, depth of discussion, and data presentation.

Specific comments:
1.    The study mentions that 69 lung tissue samples were collected in 2018, but it does not justify why older samples were used, instead of more recent ones. The authors should clarify the relevance of these samples in the current context. Were these samples the most recent available at the time of analysis, or were they selected for a specific reason? 
2.    The attempt to isolate PPV8 in cell culture was unsuccessful. This is an important limitation, and alternative approaches or future work should be suggested to address this issue. Were there any observable cytopathic effects during infection of the culture?
3.    The phylogenetic trees (Figures 4 and 5) need better visualization, including: Improved readability of strain names. Inclusion of a distance scale bar, which is currently missing.
4.    The manuscript contains grammatical errors, unclear wording, and redundant phrases, which compromise the readability. Some examples of errors and areas needing improvement:
Line 12: "And any known parvoviruses were identified, and five nearly full-length genome sequences were determined and analyzed." This sentence is unclear and redundant.
Line 34: "Currently, there are 9 genera of porcine parvovirus, of which Protoparvovirus, Tetraparvovirus, Chapparvovirus, Copiparvovirus and Bocaparvovirus infect swine."
5.    Figure 3: The title is inconsistent with the figure’s contents. Please verify and correct.
6.    Table A2: The title “The information of PPV strains used in this investigation” is incorrect. Suggested Revision: “Information on parvovirus strains used in this study.”
7.    Reference Errors: Line337-340: The references 30-31 are not related to the content.     The authors should ensure that all references are accurately cited and relevant to the discussion.
8.    Due to multiple typographical and grammatical errors, I strongly recommend professional proofreading or language editing to enhance clarity and readability. Improving linguistic clarity will also help ensure that the scientific impact of the study is conveyed effectively.

Author Response

请参阅附件。

Reviewer 2 Report

Comments and Suggestions for Authors

The authors of this study investigated the presence of the recently identified porcine parvovirus, PPV8, in Guangxi, China. A total of 69 samples were collected in 2018 from pigs exhibiting high fever or respiratory symptoms, of which 9 tested positive for PPV8. Five near-complete genome sequences were subsequently obtained. Phylogenetic analysis of the VP1 protein indicated that PPV8-H was closely related to PPV8-GDJM2021, whereas PPV8-A, B, G, and I exhibited greater genetic divergence. These findings provide insights into the genetic diversity of PPV8 and contribute to the understanding of prevalence of this emerging virus. However, there are some concerns need to be addressed.

  1. Reorganize the methods section to follow the chronological order of experimental procedures. To ensure logical coherence, sample collection and PCR amplification should take place before sequence analysis.
  1. Annealing temperatures are partially repeated in Methods and Table 1. Please expand Table 1 to include all PCR cycling parameters. The methods section should then refer only to Table 1.
  2. To improve interpretability, please annotate Figure 1 with the sizes of untranslated regions (UTRs), overlapping genes, intergenic regions, and individual genes.
  3. The results of both the amino acid and nucleic acid analyses are presented in Section 3.2. Please combine subsections 3.1 and 3.2 to improve logical flow.
  4. Lines 186-206 provide essential background on porcine parvoviruses. Relocating it to the introduction will help the reader understand the significance of the study.

Reviewer 3 Report

Comments and Suggestions for Authors

The current “Brief report” describes the detection, nearly full-length sequencing and phylogenetic analyses of five novel porcine parvovirus 8 strains from China. These data are very interesting, well documented and worth publishing.

Unfortunately, the part of “Discussion and conclusions” is incomprehensible and requires complete rewriting before publishing. For instance:

  • Paragraph 1 (lanes 186 to 196) jumps from a brief description of the eight PPV genotypes to mTOR signaling and autophagy pathways, before the next paragraph again start with the previous detection of PPV2 to PPV8.
  • Paragraph 3 (lanes 218ff) repeats the problems of sequencing the palindromic ends (result section) and the “running-out” of PPV8-G DNA of which the latter is part of the whole analyses presented in the result section. While the lack of palindromic sequences is acceptable for the analyses performed and could be omitted here (since it has been stated in the result section), the statement regarding PPV8-G opens the question, whether there is an “almost complete” well elaborated sequence available for this strain.
  • Paragraph 4 (lanes 234ff): There are no data available for the growth in culture and isolation of PPV8. In consequence this paragraph should be removed.
